# Fluorinated Molecules and Nanotechnology: Future ‘Avengers’ against the Alzheimer’s Disease?

**DOI:** 10.3390/ijms21082989

**Published:** 2020-04-23

**Authors:** Meghna Dabur, Joana A. Loureiro, Maria Carmo Pereira

**Affiliations:** LEPABE, Department of Chemical Engineering, Faculty of Engineering of the University of Porto, s/n, R. Dr. Roberto Frias, 4200-465 Porto, Portugal; meghna.dabur@outlook.com

**Keywords:** amyloid-beta peptide, BACE inhibitors, amyloid-beta aggregation, fluorine, nanoparticles

## Abstract

Alzheimer’s disease (AD) is a serious health concern, affecting millions of people globally, which leads to cognitive impairment, dementia, and inevitable death. There is still no medically accepted treatment for AD. Developing therapeutic treatments for AD is an overwhelming challenge in the medicinal field, as the exact mechanics underlying its devastating symptoms is still not completely understood. Rather than the unknown mechanism of the disease, one of the limiting factors in developing new drugs for AD is the blood–brain barrier (BBB). A combination of nanotechnology with fluorinated molecules is proposed as a promising therapeutic treatment to meet the desired pharmacokinetic/physiochemical properties for crossing the BBB passage. This paper reviews the research conducted on fluorine-containing compounds and fluorinated nanoparticles (NPs) that have been designed and tested for the inhibition of amyloid-beta (Aβ) peptide aggregation. Additionally, this study summarizes fluorinated molecules and NPs as promising agents and further future work is encouraged to be effective for the treatment of AD.

## 1. Introduction

Alzheimer’s disease (AD) is a progressive and irreversible neurodegenerative disease [1] and is responsible for about 70% of all late-onset dementia cases [2]. With the increase in research and development in medicine for heart diseases and cancer, there has been an overall increase in life expectancy. Thus, deaths reported from heart diseases and various types of cancer have decreased since 2000, whereas deaths from AD have increased to 145% [3]. This reveals the lack of therapies in the field of AD. In 2018, the annual cost of people with Alzheimer’s or dementia, including care provided by family members or caregivers was estimated to be 18.5 billion hours with the value cost at nearly $234 billion [3]. Therefore, AD is considered to be a social and economic burden in our society. 

The exact mechanism underlying AD is still debatable [4]. The most relevant pathogenic hallmarks of Alzheimer’s are associated with an alteration of the structure of the proteins amyloid-beta (Aβ) and Tau protein and their subsequent aggregation. Those aggregates deposit in the brain, extracellularly (Aβ) or appear inside the cells (aggregates of hyperphosphorylated Tau protein) [5,6,7]. Several facts have proven a causative role of the Aβ in the disease’ onset. Physiological Aβ in monomeric form is benign, but by an unknown mechanism it aggregates and becomes neurotoxic [8]. These aggregates deteriorate healthy brain cells, causing loss of memory and cognitive impairment [9]. The modern era of drug development for AD treatment has begun with cholinergic hypothesis [10,11]. Since then, only four cholinesterase inhibitors and memantine have been marketed for the delay of AD [12]. Out of which, one cholinesterase inhibitor (Tacrine) has been inactive due to its hepatotoxicity [13]. Current therapeutics available for the treatment for AD only relieve symptoms, cannot terminate/reverse the aggregation of amyloid deposits in the brain. This leads to the serious concern and urgent need for disease modifying treatments that can prevent, delay, and slow down the progression by targeting one of the most evident key pathophysiology mechanisms that cause AD [14,15].

Fluorine substitution has proven effective for therapeutic. Fluorine and fluorinated molecules have distinct properties such as low surface tension, good biocompatibility, sensitive temperature-dependent miscibility with organic solvents, and hydrophobic properties [16,17,18]. The conjugation of drug molecules with fluorine atoms have been explored but gained limited attention in the field of AD. 

However, one of the limiting factors in developing new drugs for AD is the blood–brain barrier (BBB). Pharmaceuticals show disadvantages like inability to cross the BBB [19], weak therapeutic efficiency, poor bioavailability, and toxic side-affects due to the accumulation of the drug at undesired locations. In recent years, nanotechnology-based strategies have proven effective and shown capability to overcome these drawbacks innate to BBB passage and used as the carriers of drug molecules [20,21,22,23,24,25,26]. Nanoparticles (NPs) have proven to be an excellent system for transporting drug to the specified organ with controlled drug release and minimal side effects [27]. NPs guarantee a potential therapeutic outcome of a drug candidate without modifying its physiochemical properties and enhance solubility of the drug, high stability, high specificity, and high therapeutic efficacy [27,28]. It was already reported that NPs are one of the reliable approaches for reducing the death rate in patients with AD [27]. Due to their small size, the administration of NPs can be achieved via various routes including nasal, oral, and parenteral, making nanotechnology a promising approach at present and in future [29]. NPs decorated with fluorine were already proposed for AD and demonstrated to be efficient [30,31]. 

Previously, our research group has discussed the potential effect of fluorinated molecules for AD treatment [32]. The current study reveals the powerful role of fluorine for the inhibition of anti-amyloid aggregation and suggests a new therapeutic approach of fluorine in combination with nanotechnology. This paper reviews the most recent reported evidence on fluorine-containing compounds and fluorinated NPs that have been designed and tested for the treatment of AD. The systematic literature search was conducted using PubMed, Science direct, Google Scholar, Scopus, and Web of Science as online databases until March 2020. Only papers written in English were considered with unlimited publication date.

## 2. The Role of Fluorine in the Development of Therapeutic Drugs

The chemistry and the biological applications of fluorine and fluoroorganic compounds were discussed by many research groups in the last 20 years [33,34,35,36,37,38,39]. Fluorine chemistry has gained huge attention and undergone a significant transformation from only a subfield of organic chemistry into a major area of interdisciplinary research associated with medicinal chemistry and drug discovery [28]. Nowadays, an increasing number of fluorinated drugs are available in commercial market and a significant number of fluorine-containing compounds have been approved by the FDA for medical uses [40,41].

Fluorine substitution influences the physical and chemical properties, rate of metabolism, lipophilicity, and biological activity of drugs or a bio-active compound [42]. Fluorine atoms have small atomic size (1.47 Å), high electronegativity (3.98/Pauling scale), low polarizability, and high C-F bond strength (116 kcal/mol) [43]. Figure 1 presents a schematic diagram that summarizes the physiochemical and pharmacokinetics properties of fluorine for therapeutic drugs development.

The replacement of a hydrogen atom, methyl, hydroxyl group, and other univalent atoms with a fluorine atom can alter the dipole moments, the pKa, chemical reactivity, and stability of neighboring functional group(s) of a lead compound [33]. These characteristics of fluorine improve the oral absorption in the blood and further enhance the bioavailability of drug candidates. Fluorine substitution can prevent oxidative metabolism at the site of metabolic attack as C-F bond has higher stability than the C-H bond (C-H = 99 kcal/mol) [33]. Additionally, fluorine substitution influences drug metabolism when placed at next to or far-off sites to the site of metabolic attack due to its stereo electronic properties [35]. The inclusion of fluorine in a bioactive drug candidate can enhance the interaction, binding efficacy, and selectivity towards a target protein [44]. The lipophilic behavior of fluorine atom and fluorine-containing functional groups enhance membrane penetration of drugs and facilitate crossing of molecules into the blood–brain barrier [33]. Numerous central nervous system (CNS) drugs contain either CF_3_ group or a fluoro-phenyl group which subsidizes overall pharmacological activity of the compounds [33,45].

However, unsuitable fluorine positioning in a bio-active compound or a drug molecule can cause stability and toxicity issues. The concern regarding the toxicity of fluorinated molecules has drawn attention to medicinal chemists in the last few years due to the instability of C-F bond in the presence of a nucleophile or drug-metabolizing enzymes [46]. There is a clinical evidence that shows the prolong use of antifungal voriconazole can increase plasma fluoride levels and lead to painful periostitis/exostoses in patients [47]. 

With the discovery of more innovative fluorinated molecules, medicinal chemists, and other researchers are equipped with more powerful tools to recognize the potential liability of fluorinated molecules/drugs and play around in placing fluorine atoms at different position for tackling such issues [46].

### 2.1. The Role of Fluorine-Containing Compounds in the Modulation of Amyloid-Beta Peptide

Various research groups have studied, designed, and tested fluorine-containing compounds for the inhibition of oligomerization of Aβ peptide (either Aβ_40_ or Aβ_42_). Many findings have suggested that the conversion of alpha(α)-helical structure (non-pathogenic) to beta(β)-sheet configuration (pathogenic) is responsible for aggregation [48,49]. It was demonstrated that fluorinated alcohols act as a conformational effector which can refold the β-conformation into the α-helical structure and alter the intermolecular interaction in a protein solution [50]. Vieira et al. [51] has presented the effectiveness of different fluorinated alcohols for β-to-α refolding process of Aβ_40_. Trifluoroethanol (TFE) and hexafluoroisopropanl (HFIP) (Figure 2A) were experimented in comparison with ethanol (EtOH), as well as the adsorption at fluorinated surfaces. Ethanol found to be the weakest of all the agents that were studied in terms of the hydrophobic character. HFIP was a stronger α-helix inducer than TFE; at 12% v/v_sol_, the Aβ has reduced its total β-sheet content by 50% and 80% for TFE and HFIP, respectively. The higher the fluorination rate, the greater the efficiency. The molecules of fluorinated alcohol can enclose the protein molecules, easily infiltrate into the aggregates and detach them [51].

Another strategy was explored, like the conjugation of fluorinated molecules into peptides to modify the proteolytic stability [52,53], hydrophobicity [54], and to control the conformations of a peptide chain [55]. Loureiro et al. [56] synthesized fluorinated peptides with a sequence of five amino acids against Aβ aggregation. These fluorinated peptides were based on the central hydrophobic residues of Aβ and in the iAβ_5_ β-sheet breaker peptide; LVFFD sequence. In this sequence, valine was substituted by either 4,4,4-trifluorovaline or 4-fluoroproline or 3,4,5-trifluorophenylalanine (Figure 2B). The thioflavin T (ThT) binding assay showed that two of the fluorinated peptides (LVFfFD–PEG and LVfFFD–PEG) are able to delay the aggregation of the Aβ_42_ by 1.5 h and 3.1 h. The authors proposed that the position of the fluorine at the extremity of the conjugate seems to influence the hydrophobicity of the sequence. Botz et al. [57] synthesized two peptides, one containing the trifluoromethylated (Tfm) analogue (R)-α-trifluoromethylalanine (Figure 2C) and the other incorporating the nonfluorinated analogue α-aminoisobutyric (Aib). The in vitro studies of these two peptides were evaluated and compared using fluorescence spectroscopy (thought the ThT assay), circular dichroism (CD), and liquid state nuclear magnetic resonance (NMR) for the inhibition of Aβ aggregation. The ThT assay studies of fluorinated amino acids have shown the declination of lag phase of Aβ_42_ from 10 h to 15 h with a reduction in final fluorescence intensity by a factor of 2 and decrease in aggregation rate. The ThT assay results were coherent with the transmission electron microscopy (TEM) results as smaller and thinner fibrils were formed in the presence of a fluorinated amino acid while long and twisted amyloid fibrils with widths between 10 and 15 nm were observed in the presence of a non-fluorinated amino acid analogue. The CD signal of Aβ_42_ in the presence of a fluorinated peptide remained stable for at least 96 h, showing no evidence of β-sheet structure and no conformational transition was observed, while in case of a non-fluorinated peptide transition occurred at 50 h. ^1^H-NMR studies were performed to study the kinetics of monomeric soluble Aβ_42_ in the presence and absence of a fluorinated amino acid. The complete disappearance of monomeric NMR signals was observed after 14 h in case of a non-fluorinated peptide. Whereas in case of a fluorinated amino, acid, monomeric Aβ_42_ was detected even after 20 h incubation. These results advocated that the Tfm group is more effective and has a better affinity with Aβ peptide than the methyl groups of the Aib amino acid. 

Török et al. introduced organofluorine Aβ_40_ fibrillogenesis inhibitors. The core structure containing indol-3-yl, trifluoromethyl, hydroxyl, and carboxylic acid ester functions in the chain (Figure 2D) [58]. Trifluoro-hydroxylindolyl-propionic acid esters performed well in dose-dependent studies, three of these compounds (Figure 2D) practically exhibited complete blocking of fibril formation (percentile values are between 93% and 96%) in favorable (mol_inhibitor_/mol_peptide_ = 10) stoichiometry. The ThT fluorescence and TEM results were consistent and showed a network of long fibrils in an inhibitor-free Aβ sample while no aggregates formation was observed in organofluorine inhibitors Aβ sample. The ^19^F-NMR studies indicated the interactions between organofluorine inhibitors and the Aβ peptide. For understanding the nature of this interaction, the data showed that the removal of CF_3_ groups generated a substantial change in activity; these molecules rather acted as fibrillogenesis promoters. The in vitro studies interpreted that CF_3_ is responsible for activating OH, which binds to the peptide (possibly to one or both lysine residues, 16 and 28 of Aβ sequence). Therefore, CF_3_ plays a fundamental role in inhibiting the amyloid fibrillation process. Additionally, Sood et al. [59] studied effective organofluorine-chiral inhibitors [58] and their enantiomeric compounds for their potency in in vitro. They have confirmed that the individual enantiomers of substituted trifluoro-hydroxylindolyl-propionic acid esters are potent inhibitors of the Aβ self-assembly, similar to their racemic mixtures. This research data suggested that the inhibitory effect is not a result of highly stereospecific binding between inhibitors and the peptides, but rather non-stereospecific binding forces (chemical nature, substituents, and electronic character of the molecule). Later, Sood et al. [60] observed that 5′-halogen substituted 3,3,3-trifluoro-hydroxylindolyl-propionic acid esters exhibited potential to work as disassembly agents against the preformed fibrils, irrespective of the stereo specificity associated with each compound in reference with melatonin [61] (a well-known amyloid inhibitor). The data followed by Fourier-transform infrared spectroscopy (FTIR) showed a shift from 1630 cm^−1^ (characteristic of β-sheet structure) to around 1645–1650 cm^−1^ suggested the formation of oligomers or other less ordered species [60]. Therefore, organofluorine molecules were classified as ‘dual nature inhibitors’ since they were good fibrillogenesis inhibitors [58,59] and disassembly agents [60] as well. This dual action makes them potential candidates for drug development of AD treatment in future. 

Another strategy replacing univalent atom or hydroxyl group with fluorine isotopes were proposed as both diagnostic and therapeutic. Elmaleh et al. [62] claimed Cromolyn-based (anti-inflammatory agent) radiolabeled (isotopes of ^13^C, F^18^ or F^19^ label) derivatives. The fluorinated derivatives (Figure 2E) were proposed as both AD diagnostic and therapeutic agents; biomarkers (F^19^ aided magnetic resonance imaging) to monitor the advancement of the disease and the inhibitors (F^18^ facilitated positron emission tomography: PET) of Aβ peptide aggregation. The in vivo bio-distribution of fluorinated Cromolyn (F^18^) derivatives showed a substantial brain uptake and clearance activity. Additionally, the oligomerization of Aβ was decelerated by ^18^F-ligand derivative by approximately 2.5 times. Reed et al. [63] invented small molecules comprising a biaryl-scaffold with a nitro and a benzyl amino moiety (Figure 2F). These compounds regarded as good anti-aggregating agent and metabolically stable, with low IC_50_ value against Aβ peptide concentration via ThT assay. The organofluorine analogue was found to be a good BBB penetrable compound exhibiting remarkable blood and brain exposure levels (brain/plasma ratio: 1.54). The in vivo efficacy trials on 5 × FAD (Familial Alzheimer’s disease) transgenic AD mouse indicated that organofluorine inhibitor has beneficial effects on their cognitive impairment. Horwell et al. [64] discovered a series of small compounds with potential anti-AD properties. The compound bearing 2-fluorobenzonitrile moiety surfaced as a promising candidate against AD with a remarkable anti-aggregating Aβ_42_ capability and cellular activity (Figure 2G). The analogues with a carboxyl or an amide substituent were found inactive, highlighting the biological activity of 2-fluorobenzonitrile framework. The in vitro experiment on rat brain cell lines showed that compound has blocked the toxic effect on long-term potentiation (LTP). Moreover, the same research group [65] has also synthesized a compound bearing 8-fluoro-3,4-dihydro-2Hbenzo [1,4] oxazine moiety that presented an excellent neuroprotective profile in in vitro against amyloid toxicity (Figure 2H). Additionally, it blocked the toxic effect in rat hippocampal brain slices in the presence of Aβ_42_ in LTP.

Another strategy to understand the role of fluorine for the modulation of the Aβ peptide was addressed by Giacomelli et al. [66]. They studied the effect of a hydrophobic polytetrafluoroethylene (Teflon) (Figure 2I) as a solid sorbent surface on the conformation of Aβ_40_. The in vitro experiments were carried out by dissolving Aβ_40_ (containing monomers, small oligomers, and fibrillary aggregates) at different pH (4, 7, and 10). Upon analysis of peptide-surface (hydrophobic Teflon) interaction using CD, the random structure (pH 10) and the β-sheets (pH 7) were converted into α-helical structures. However, when the fluorinated surface becomes crowded, lateral interactions amongst the adsorbed peptide molecules persuades β-sheet structure formation. These findings showed that the fluorinated surfaces strongly promote α-helix reformation.

Additionally, theoretical studies were done with the fluorinated compounds against AD. Török et al. reviewed the basis of 321 compounds, discussed the role of each chemical structure and the effect of their substituents in designing the map of their inhibition against Aβ peptide [67]. Based on their analysis, acidic moieties like CF_3_, appeared to be the second most important substituent of these small organofluorine inhibitors implying fluorinated motifs as possible binding units to Aβ peptide. Khosravan et al. [68] theoretically investigated the effect of fluorine substitution on an anti-Alzheimer drug donepezil (FDA approved), including seven fluorinated derivatives of it, via density functional theory (DFT) and molecular docking calculations. This study shows that the fluorine substitution influences the solubility, molecular polarity, and structural stability of the donepezil drug. After analyzing natural bond orbital (NBO) and atomic force microscopy (AFM) calculations, they indicated that the fluorination of donepezil seems to enhance intramolecular hydrogen bonding, charge distribution, and aromaticity of donepezil. 

Table 1 summarizes the aforementioned results already discussed in brief.

### 2.2. The Role of Fluorine-Containing Compounds in the Modulation of the Secretases

β-Site amyloid precursor protein cleaving enzyme (BACE1) plays a crucial role in controlling the formation of Aβ peptide as it is the only enzyme responsible for the β-secretase activity in the brain [69]. Therefore, BACE1 inhibitors present the possibility of disease-modifying treatment for AD. Since 1999, after the identification of the potential pharmacological target along with the results from the BACE1 knockout mice [70], many research groups and companies have invested in developing BACE1 inhibitors (Table 2). Several companies like Pfizer, Bristol-Meyers Squibb (BMS), Lilly, Roche, Novartis, etc. have introduced fluorine atom and fluoro-methyl substituents to the BACE1 inhibitors to increase potency, improve cellular activity and metabolic stability. We are presenting selected BACE1 inhibitors containing at least one fluorine element in the chemical structure that have been designed and tested between 2010 and 2020 [69]. 

Inspired from the work by Elan and Pfizer [71], Fustero et al. [72] synthesized fluorinated ethanolamines (Figure 3A) to analyze the essential fragments for the stereo-selective synthesis of hydroethyl secondary amine (HEA). They substituted phenyldifluoromethyl at the α-carbon of the HEA and explored the chemical space of the inhibitor by replacing hydrogen atoms at the benzylic position by fluorine atoms for enhancing the pharmacological profile of the series [44,73,74]. The biological evaluation of these derivatives disclosed a notable BACE1 inhibitor activity. Docking studies showed the potential of fluorine atoms in influencing the potency of the inhibitors [72].

In 2015, Lilly’s LY-2886721 [75] (Figure 3B) was the first 1,3-thiazine based BACE1 inhibitor advancing to a phase 2 trial. In the phase 1 trial, LY-2886721 was found efficient and decreased the Aβ levels in cerebrospinal fluid (CSF). However, this inhibitor was terminated by the company due to liver toxicity. Later, they reported a modification of LY-2886721 by introducing a fluorine atom at the ring junction and over all enhancing the basicity of the compound. The fluorinated analogue 4a-Fluoro-furo[3,4-d][1,3]thiazine [76] (Figure 3C) showed improvement in biological parameters and potent reduction of amyloid levels in in vitro and in vivo as compared to LY-2886721. Similar to the 1, 3-thiazine analogue, Lilly patented several fluorinated compounds [77,78,79], using difluoroethane group (CF_2_CH_3_) on the tetrahydrofuran (THF) ring. These fluorinated analogues have shown good BACE1/2 selectivity and a slight improvement in BACE1 potency. Lilly claimed another THF fused-1, 3-oxazine BACE1 inhibitor [78] in the same patent application with the incorporation of a fluorine atom. This fluorinated inhibitor (Figure 3D) showed high BACE1/2 selectivity and that can be contributed to the introduction of bulky difluoroethyl group on the THF ring which is in close vicinity to the flap region. The in vivo experiments showed that the fluorinated inhibitor at 24 h and 48 h post dose reduced Aβ levels 65% and 55% in CSF.

In THF-fused thiazine series, Eisai [80,81] has further modified THF substituents exploring substituents from CH_3_, CH_2_F, CHF_2_, to CF_3_ (Figure 3E). This was carried out to adjust pKa and selectivity over BACE2 for both in in vitro and in vivo experiments. Shingoi and Janssen [82] together claimed a series of patent focusing on 1,3-thiazine based BACE1 inhibitors. These 1, 3-thiazine inhibitors were decorated with fluoro(alkyl) substituents like, CHF_2_, CF_3_, and CF_2_CH_3_ (Figure 3F,G,H, respectively) at the 6^th^ position for improving the biochemical properties. These series of inhibitors found to be non-P-gp (non-polyglycoprotein) substrates but has shown less selectivity for BACE1 over BACE2.

Multiple companies like Roche [83], Novartis [84], Amgen [85], and Shionogi [86] have also explored fluorine, fluoromethyl, difluoromethyl, or trifluoromethyl with 1,3-oxazine based amidines as BACE1 inhibitors for the treatment of AD. Roche’s publication on fluorinated 1,3-oxazines (Figure 3I) discusses not only the influence of degree of fluorination to modify basicity (suitable range: pKa 5.8–7.4), but other parameters including cellular and enzymatic activity, P-gp efflux in in vitro and in vivo potency [87]. The study revealed that the CF_3_ substituted oxazine was highly potent and excellent brain penetrant BACE1 inhibitor and reduced significant CSF levels (both Aβ_40_ and Aβ_42_) in rats at low oral doses.

## 3. Nanoparticles Designed with Fluorine Molecules for the Treatment of Alzheimer’s Disease

Based on fluorine advantageous properties to treat AD, few research groups have tried another approach. They have developed NPs designed with fluorine molecules (Table 3). Rocha et al. [30] studied the influence of fluorinated and hydrogenated NPs on the fibrillogenesis of Aβ_40_ using CD and TEM. The fluorinated complexes designed by the group shown the inhibition of amyloid fibril formation by binding to Aβ_40_. After titrating Aβ_40_ with P_1_F complexes [Poly(N,N′-diallyl-N,N′dimethylammoniumalt-maleamic Carboxylate) as P_1_ complexed with perfluorododecanoic acid (F)] the CD spectra showed the content of α-helix to be 8%, 16%, and 31% after the addition of 2, 4 and 8 g L^-1^ of P_1_F complexes. The unordered content decreased to 27% where β-sheet structures remained almost constant. While in case of P_1_H [Poly(N,N′-diallyl-N,N′dimethylammoniumalt-maleamic Carboxylate) as P_1_ complexed with dodecanoic acid (H)], the hydrogenated analogues did not seem to induce any α-helix structure on Aβ_40_. The fraction of β-sheet increased from 27% to 37% and the unordered content decrease from 56% to 37%. After the addition of fluorinated NPs P_2_F [(Poly(N,N′-diallyl-N,N′-dimethylammonium-alt-N-phenylmaleamic Carboxylate as P_2_ complexed with (F)], the α-content was, however smaller as compared to P_1_F (from 3% to 21% at 8 g L^−1^). This was explained through the steric hindrance caused by the other phenyl groups and inhibited the interaction of fluorine molecules with the α-helical structure. TEM images have shown the interaction of fluorinated NPs with Aβ aggregation. This is due to the high negative zeta potential of fluorinated NPs and their hydrophobicity that have shown significant inhibition towards the Aβ formation. Following that, Saraiva et al. [31] studied the effect of similar fluorinated and hydrogenated NPs on Aβ_42_ oligomerization and cytotoxicity. The data from CD, AFM, immunoblotting, and SDS-PAGE were consistent and indicated that fluorinated NPs promote β-to-α conformational transition, delayed the oligomerization of Aβ_42_ and hence lower the peptide-induced cytotoxicity. Whereas the hydrogenated NPs have promoted the nucleation and aggregation rate of Aβ_42_ monomers and oligomers. This might be due to the intermolecular interaction between fluorinated NPs and two hydrophobic residues responsible for Aβ oligomerization (Leu17-Ala21 and Ala30-Ala42) or with the residues lying in proximity of Aβ assembly, preventing the inter-peptide hydrophobic interactions and intervening the process of further aggregation. The caspase-3 (an indicator of apoptosis) fluorometric assays indicated that fluorinated NPs produced the caspase-3 activation values that are almost similar to the values without Aβ_42_ peptide (1.1 ± 0.07 fluorescence intensity per mg of protein for the control without peptide). Whereas enzymatic activity was increased in the presence of hydrogenated NPs (1.9 ± 0.15 of fluorescence intensity per mg of protein). Their research study reported that fluorinated NPs exhibits anti-oligomeric and anti-apoptotic activity and has also increased cell viability in the presence of Aβ_42_ oligomerization.

## 4. Conclusion and Future Perspective

In the last two decades, the fluorine-containing compounds have attracted a great deal of academic and industrial interest owing to the overall remarkable properties like high metabolic stability, distribution, and excretion properties etc. From this research study, we have reviewed the strategies that have been explored using fluorinated molecules for the treatment of AD. Fluorinated alcohols were studied in regard to protein misfolding. This study showed the effect of CF_3_ group on the conformational change (β-to-α conversion) and has opened the possibility for the therapeutic in vivo use of fluorinated drugs to prevent Aβ aggregation. In consistent with this, adsorption studies of Aβ solutions on Teflon surface have strongly promoted re-formation of α-helix. Few research groups have also explored fluorinated amino acids and their incorporation into peptides, revealing fluorinated peptides as effective β-sheet breaker peptides. So, the modification of compounds/peptides/anti-amyloid inhibitors with fluorine contributes in understanding the interaction between Aβ peptide and fluorinated molecules. The fluorination increases the hydrophobicity of the molecules and interact with the hydrophobic region of the Aβ peptide. Fluoro(methyl) substituents are not only limited to one class of anti-amyloid agent but have also been explored with BACE1 inhibitors. Big Pharmaceuticals like Novartis, Lilly, and Roche have introduced fluorine into the chemical structure of many patented BACE1 inhibitors to improve overall biochemical and pharmacological properties of the lead compound. 

These findings reveal the power of fluorine and have proven it to be a ‘potential superhero’ in the therapeutic treatment of AD.

The currently available therapeutic treatments for AD are often compromised due to BBB passage. Many promising Aβ ligands or organofluorine compounds that have been discussed in literature review cannot cross the BBB junction. A promising idea is to explore a medically accepted vehicle (for example: nanocarriers/particles) in combination with Aβ ligands or fluorinated molecules to cross the BBB. NPs surface can easily be modified with specific molecules or antibodies to reach site-specific organ, avoid recognition by the macrophages of the reticuloendothelial system (RES) and successful penetration into the BBB. Only a few researchers have studied NPs decorated with fluorine for the treatment of AD. Fluorinated NPs exhibit anti-oligomeric and anti-apoptotic activity and are demonstrated to be effective. These findings will generate more investigations in the chemistry of fluorinated NPs and could lead to the development of potential nano-fluorinated vehicles for Alzheimer’s disease. 

The real culprit behind AD is still unidentified and the current therapeutic treatments lack therapeutic efficacy to attenuate the symptoms of this disease. Keeping this in mind, new innovative therapeutic approaches are in demand. Therefore, we suggest a combination of nanotechnology with fluorinated molecules can be a promising/potential therapeutic treatment to meet the desired pharmacokinetic/physiochemical properties for crossing the BBB passage and treat AD. 

## Figures and Tables

**Figure 1 ijms-21-02989-f001:**
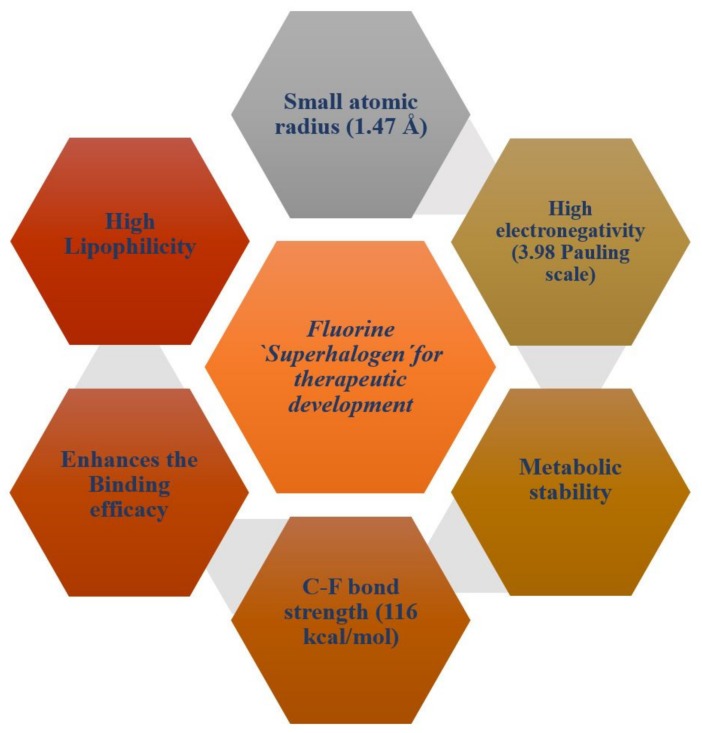
A schematic diagram showing physiochemical and pharmacokinetics properties of fluorine for therapeutic drugs development.

**Figure 2 ijms-21-02989-f002:**
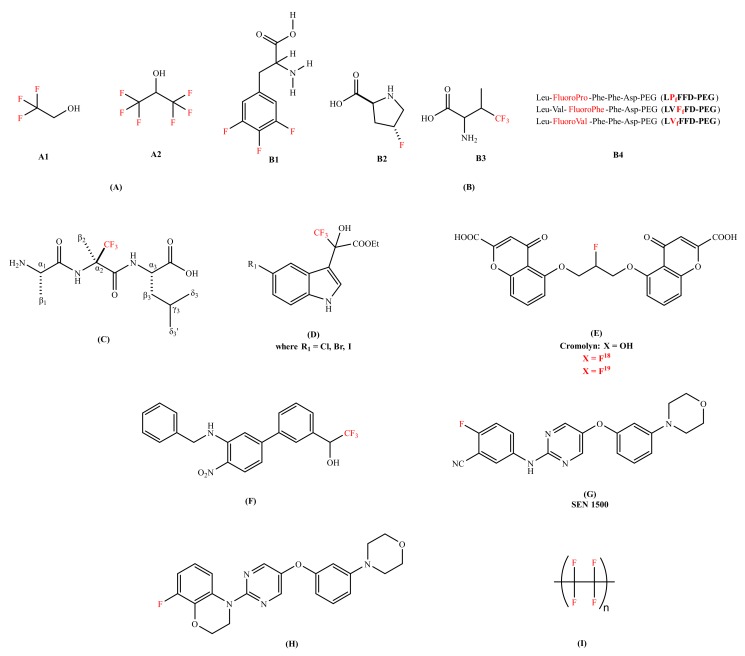
Chemical structures of the fluorine-containing compounds for the inhibition of amyloid-beta peptide. (**A**) Fluorinated alcohols (A1: 2,2,2-trifluoroethan-1-ol (TFE), A2: 1,1,1,3,3,3-hexafluoropropan-2-ol (HFIP));. (**B**) Fluorinated amino acids and peptides (B1-3,4,5-trifluorophenylalanine, B2- 4-fluoroproline, B3- 4,4,4-trifluorovaline, B4- Fluorinated peptides conjugated to Polyethylene glycol (PEG)); (**C**) (R)-α-trifluoromethylalanine peptide (H-Ala-(R)-Tfm-Ala-Leu-OH); (**D**) Trifluoro-hydroxylindolyl-propionic acid esters;. (**E**) Cromoyln-based derivative (5,5’-((2-fluoropropane-1,3-diyl)bis(oxy))bis(4-oxo-4H-chromene-2-carboxylic acid)); (**F**) Organofluorine inhibitor (1-(3’-(benzylamino)-4’-nitro-[1,1’-biphenyl]-3-yl)-2,2,2-trifluoroethan-1-ol);.(**G**) SEN1500 (2-fluoro-5-((5-(3-morpholinophenoxy)pyrimidin-2-yl)amino)benzonitrile); (**H**) 8-fluoro-3,4-dihydro-2H benzo [1,4] oxazine inhibitor (8-fluoro-4-(5-(3-morpholinophenoxy)pyrimidin-2-yl)-3,4-dihydro-2H-benzo[b][1,4]oxazine); and (**I**) Poly(1,1,2,2-tetrafluoroethylene) (Teflon). In red color are represented fluorine molecules (F) and trifluoromethyl groups (CF_3_).

**Figure 3 ijms-21-02989-f003:**
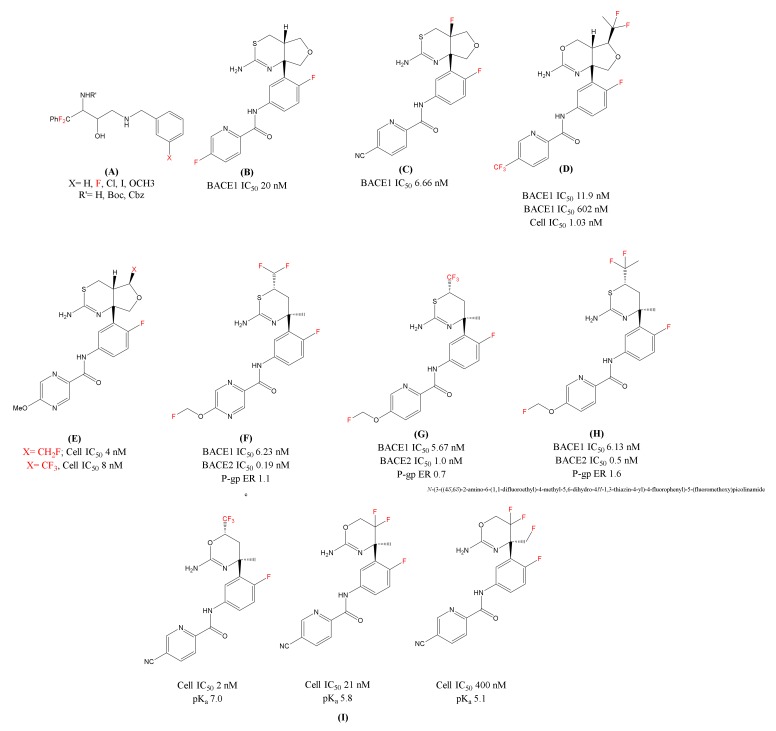
Chemical structures of the fluorine-decorated BACE1 inhibitors (**A**) Fluorinated ethanolamines; (**B**)LY-2886721(N-(3-((4aS,7aR)-2-amino-4a,5-dihydro-4H-furo[3,4-d][1,3]thiazin-7a(7H)-yl)-4-fluorophenyl)-5-fluoropicolinamide); (**C**) Fluorinated analogue of LY-2886721(N-(3-((4aR,7aR)-2-amino- 4a-fluoro-4a,5-dihydro-4H-furo[3,4-d][1,3]thiazin-7a(7H)-yl)-4-fluorophenyl)-5-cyanopicolinamide); (**D**) Lilly’s Fluorinated Inhibitor (N-(3-((4aR,5S,7aR)-2-amino-5-(1,1-difluoroethyl)-4a,5-dihydro-4H-furo[3,4-d][1,3]oxazin-7a(7H)-yl)-4-fluorophenyl)-5-(trifluoromethyl)picolinamide); (**E**): Eisai’s BACE1 inhibitor [1,3] thiazine series Fluoro(methyl) analogues; F, G and H) Shingoi and Janssen’s organofluorine substituted BACE1 inhibitors; (**F**) N-(3-((4S,6S)-2-amino-6-(difluoromethyl)-4-methyl-5,6-dihydro-4H-1,3-thiazin-4-yl)-4-fluorophenyl)-5-(fluoromethoxy)pyrazine-2-carboxamide; (**G**) N-(3-((4S,6S)-2-amino-4-methyl-6-(trifluoromethyl)-5,6-dihydro-4H-1,3-thiazin-4-yl)-4-fluorophenyl)-5-(fluoromethoxy)picolinamide; (**H**) N-(3-((4S,6S)-2-amino-6-(1,1-difluoroethyl)-4-methyl-5,6-dihydro-4H-1,3-thiazin-4-yl)-4-fluorophenyl)-5-(fluoromethoxy)picolinamide); and (**I**) Roche’s Fluorinated 1,3-oxazines inhibitors.In red color are represented fluorine molecules (F) and trifluoromethyl groups (CF_3_).

**Table 1 ijms-21-02989-t001:** Fluorinated molecules and surfaces developed for the inhibition of amyloid-beta peptide.

Compounds	Site of Action	Observed Effects	Models Used	Reference
Fluorinated alcohols (TFE and HFIP)	Aβ_40_ peptide	Induce conformational transition from β-sheet to α-helical structure	Stock solution of Aβ_40_ peptide	[51]
LVfFFD-PEG and LVFfFD-PEG	Aβ_42_ peptide	Delay the formation of the Aβ aggregates	Stock solution of Aβ_42_ peptide	[56]
(R)-α-trifluoromethylalanine		Delays the conformational transition for at least 96 h; Slows the kinetic depletion rate of Aβ monomers	Stock solution of Aβ_42_ peptide	[57]
5′-halogen substituted 3,3,3-trifluoromethyl-2-hydroxyl-(indol-3-yl)-propionic acid esters	Aβ_40_ peptide	Inhibits the fibrillogenesis; Disassemble preformed fibrils	Stock solution of Aβ_40_ peptide	[58,59,60]
Cromolyn-based fluorinated derivative	Aβ_40_ and Aβ_42_ aggregates	Inhibits the Aβ oligomerization; Displays significant brain uptake and clearance activity	Stock solution of Aβ_40_ and Aβ_42_ peptideand WT mice	[62]
Organofluorine inhibitor	Aβ_40_ peptide	Improve the cognitive impairment; Enhance the BBB permeability	Stock solution of Aβ_40_ peptide and mice	[63]
SEN 1500	Aβ_42_ peptide	Demonstrates anti-aggregating capability; Blocks the toxic effect in LTP	Stock solution of Aβ_42_ peptide, SH-SY5Y cells and 7PA2 CM cells	[65]
8-fluoro-3,4-dihydro-2H benzo [1,4] oxazine inhibitor		Inhibits Aβ aggregation; Shows excellent neuroprotective profile	Stock solution of Aβ_42_ peptide, SH-SY5Y cells, hippocampal slices of male young rat (6–8 weeks old)	[65]
Fluorinated surface (Teflon)	Aβ_40_ peptide	Promotes α-helix reformation	Stock solution of Aβ_40_ peptide	[66]

**Table 2 ijms-21-02989-t002:** Fluorinated BACE1 inhibitors for the Alzheimer’s disease (AD) treatment.

Compounds	Site of Action	Observed Effects	Model Used	Reference
Fluorinated ethanolamines	β-secretase (BACE1)	Inhibits BACE1 activity	Enzymatic assay (human BACE1), human neuroblastoma SKNBE2 cells	[72]
LY-2886721		Decreases the Aβ levels in CSF	Human. Terminated after phase 2 due to liver toxicity	[75]
Fluorinated LY-2886721		Reduces the amyloid levels	HEK293 cells (Human BACE1)PDAPP young mice	[76]
1,3 oxazine-based BACE1 inhibitor (difluoroethyl substituted analogue)		Display good BACE1/2 selectivity; Reduce Aβ levels in CSF	HEK293 cells (Both human BACE1 and BACE2) male beagle dogs	[78]
Eisai’s BACE1 inhibitor [1,3] thiazine series Fluoro(methyl) analogues		Enhance the basicity and show selectivity over BACE2	Human/Rat Aβ_42_; neuronal cultures of rat’s fetus brain	[80,81]
Organofluorine substituted BACE1 inhibitors		Improve the drug efficacy (non-P-gp substrates)	Neuroblastoma SH-SY5Y cells, human liver microsomes, ICR mice (7–9 weeks old)	[82]
Fluorinated oxazines analogues		Enhance potency and basicity; Reduce the Aβ levels at low doses	Enzymatic assays (BACE1 and BACE2), HEK293 cells, LLC-PK1 cells, female WT-mice	[88]

**Table 3 ijms-21-02989-t003:** Fluorinated nanoparticles used for the treatment for AD.

Compounds	Site of Action	Observed Effects	Model Used	Reference
Fluorinated nanoparticles	Aβ_40_	Inhibition of amyloid fibril formation	Stock solution of Aβ_40_ peptide (in vitro)	[30]
	Aβ_42_	Anti-oligomeric and anti-apoptotic activity	Stock solution of Aβ_42_ peptide (in vitro) and SH-SY5Y cells (in vitro)	[31]

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
