# Peer review of "Fluorinated Molecules and Nanotechnology: Future ‘Avengers’ against the Alzheimer’s Disease?"

_ijms, 2020, doi:10.3390/ijms21082989_

Round 1
Reviewer 1 Report
The review article summarizes the most recent treatment methods for Alzheimer’s disease using fluorine-containing compounds and fluorinated nanoparticles. The authors suggest that the latter would effectively cross the blood-brain-barrier and reach the desired pharmacokinetic/physiochemical properties. Hence, it is a promising method to treat Alzheimer’s disease. The manuscript is interesting to read. However, I suggest following minor changes.
- Terms “in vivo” and “in vitro” should be written in italic font throughout the manuscript.
- Line 53-55: “In the recent years, nanotechnology-based strategies have proven effective and shown capability to overcome these drawbacks innate to BBB passage and used as the carriers of drug molecules”
I suggest to add more references to above statement.
- Line 61: “including nasal, oral, parenteral making nanotechnology…”.
Add “and” after “oral”
- Line 117: “iAβ5 β-sheet breaker peptide; LVFFD sequence (Figure 2B)”
The peptide is not shown in Figure 2B.
- Line 260: “inhibitor at 24 and 48 h post dose reduced Aβ levels 65 and 55% in CSF.”
Add “h” after 24 and “%” after 65.
Author Response
We acknowledge and appreciate the reviewer comments and we thank the reviewer for the positive opinion about our manuscript. We totally agree with the reviewer suggestions and the manuscript was carefully revised accordingly.
- Terms “in vivo” and “in vitro” should be written in italic font throughout the manuscript.
Updated.
- Line 53-55: “In the recent years, nanotechnology-based strategies have proven effective and shown capability to overcome these drawbacks innate to BBB passage and used as the carriers of drug molecules”
I suggest to add more references to above statement.
Thanks for the suggestion. More references to support the sentence were added.
Please see the references 21 to 26.
- Joshi, S. A.; Chavhan, S. S.; Sawant, K. K., Rivastigmine-loaded PLGA and PBCA nanoparticles: Preparation, optimization, characterization, in vitro and pharmacodynamic studies. European Journal of Pharmaceutics and Biopharmaceutics 2010, 76, (2), 189-199.
- Sánchez-López, E.; Ettcheto, M.; Egea, M. A.; Espina, M.; Cano, A.; Calpena, A. C.; Camins, A.; Carmona, N.; Silva, A. M.; Souto, E. B.; García, M. L., Memantine loaded PLGA PEGylated nanoparticles for Alzheimer’s disease: in vitro and in vivo characterization. Journal of Nanobiotechnology 2018, 16, (1), 32.
- Wilson, B.; Samanta, M. K.; Santhi, K.; Kumar, K. P. S.; Paramakrishnan, N.; Suresh, B., Poly(n-butylcyanoacrylate) nanoparticles coated with polysorbate 80 for the targeted delivery of rivastigmine into the brain to treat Alzheimer's disease. Brain Research 2008, 1200, 159-168.
- Brambilla, D.; Le Droumaguet, B.; Nicolas, J.; Hashemi, S. H.; Wu, L. P.; Moghimi, S. M.; Couvreur, P.; Andrieux, K., Nanotechnologies for Alzheimer's disease: diagnosis, therapy, and safety issues. Nanomedicine : nanotechnology, biology, and medicine 2011, 7, (5), 521-40.
- Yang, Z.; Zhang, Y.; Yang, Y.; Sun, L.; Han, D.; Li, H.; Wang, C., Pharmacological and toxicological target organelles and safe use of single-walled carbon nanotubes as drug carriers in treating Alzheimer disease. Nanomedicine: Nanotechnology, Biology, and Medicine 2010, 6, (3), 427-441.
- Kreuter, J., Nanoparticulate systems for brain delivery of drugs. Advanced Drug Delivery Reviews 2001, 47, (1), 65-81.
- Line 61: “including nasal, oral, parenteral making nanotechnology…”.
Add “and” after “oral”
Updated.
- Line 117: “iAβ5 β-sheet breaker peptide; LVFFD sequence (Figure 2B)”
The peptide is not shown in Figure 2B.
Thanks for noticing. Chemical structure of the peptide sequence in Figure 2B was updated.
- Line 260: “inhibitor at 24 and 48 h post dose reduced Aβ levels 65 and 55% in CSF.”
Add “h” after 24 and “%” after 65.
Updated.
Reviewer 2 Report
The review paper entitled “Fluorinated Molecules and Nanotechnology: Future 2 ‘Avengers’ Against the Alzheimer’s Disease?” by Meghna et al. summarizes the role of fluorinated molecules and nanotechnology in treating Alzheimer’s disease, with a focus on the fluorinated molecules. Though this article is informative, I have the following major concerns for this study.
- It seems to me that this article shares similarity with the published paper “ The potential effect of fluorinated compounds in the treatment of Alzheimer’s disease”, what is the advances of this study? Please highlight it in the abstract and discussion.
- On page 2, it is stated, “It was already reported that NPs are one of the reliable approach for reducing the death rate in patients with AD [22].”, however, the citation 22 seems not to focus on NPs. Similarly, “NPs decorated with fluorine were already proposed for AD and demonstrated to be efficient [24-26].” Citation 26 is about the monoclonal antibody. Please check all the citations carefully.
- The pathological factors involved in Alzheimer’s disease include more than those introduced in the introduction section, and I would suggest briefly introduce them. Furthermore, what’s the application of fluorinated molecules for tau?
- The writing style needs improvement. For example, “The complete disappearance of monomeric NMR signals were observed after 14 h in case of non-fluorinated peptide whereas the peak intensity did not decrease until 0 in presence of fluorinated amino acid indicated the presence of monomeric Aβ42 even after 20 h of incubation and slow kinetic depletion rate.” Such a sentence is hard to understand.
- Please proofread the paper and check grammar issues.
- Table 1, the authors list “therapeutic effect”, but the content is hard to understand, and I think “used as a model to understand the…” is not a therapeutic effect. I would also suggest removing the column “experiment”; for the column “models used”, it is hard to understand when the authors just put “ Aβ42 peptide sequence” or “MDRI-MDCK cells”. Similar questions applied to Table 2. Furthermore, the format should be consistent. In table 2, the authors sometimes use present tense, such as “inhibit BACE1 activity”, sometimes used past tense, such as “ Decreased the Aβ levels in CSF”, and sometimes use nouns “Potent reduction of amyloid levels”, please keep consistent.
- I suggest the authors discuss the safety and toxicity of fluorinated molecules and nanotechnology when applied alone or in combination for treating Alzheimer’s disease.
Author Response
We acknowledge and appreciate the reviewer suggestions. The modifications done based in the reviewers recommendations were highlighted in orange in the manuscript. We believe that the observations added to the text are now in agreement with the considerations made. A more detailed explanation of the alterations made is subsequently described.
Comment 1:
It seems to me that this article shares similarity with the published paper “The potential effect of fluorinated compounds in the treatment of Alzheimer’s disease”, what is the advances of this study? Please highlight it in the abstract and discussion.
Our work is in the line with the previous work done by the group. However in the present manuscript, the focus lies in showing fluorine-containing compounds as a promising anti-amyloid agent. Additionally, a different therapeutic approach including the combination with nanotechnology is proposed which was not shown in the other script. Hence the role of fluorinated particles was highlighted in a separate section (section 3).
Taking your suggestion in mind, we added a description of this other review study in the present manuscript.
Line 68 to 71: “Previously, our research group has discussed the potential effect of fluorinated molecules for AD treatment [32]. The current study reveals the powerful role of fluorine for the inhibition of anti-amyloid aggregation and suggests a new therapeutic approach of fluorine in combination with nanotechnology.”
Comment 2:
On page 2, it is stated, “It was already reported that NPs are one of the reliable approach for reducing the death rate in patients with AD [22].”, however, the citation 22 seems not to focus on NPs. Similarly, “NPs decorated with fluorine were already proposed for AD and demonstrated to be efficient [24-26].” Citation 26 is about the monoclonal antibody. Please check all the citations carefully.
Thanks for noticing. The correct citation is “Arya, M. A.; Kumar, M.; Sabitha, M.; Menon, K.; Nair, S., Nanotechnology approaches for enhanced CNS delivery in treating Alzheimer's disease. Journal of Drug Delivery Science and Technology 2019, 51.)”. It is one of the previously mentioned reference with number 20. Also, the reference 26 was removed. Maybe it was a mistake during the citations formatting. All the citations were carefully revised.
Comment 3:
The pathological factors involved in Alzheimer’s disease include more than those introduced in the introduction section, and I would suggest briefly introduce them. Furthermore, what’s the application of fluorinated molecules for tau?
We agree with the referee’s suggestion of adding more information about the disease in the Introduction Section. We therefore added this information to the manuscript, as follows:
Lines 34-40:
“The most relevant pathogenic hallmarks of Alzheimer’s are associated with an alteration of the structure of the proteins amyloid-beta (Aβ) and Tau protein and their subsequent aggregation. Those aggregates deposit in the brain, extracellularly (Aβ) or appear inside the cells (aggregates of hyperphosphorylated Tau protein) [5-7]. Several facts have proven a causative role of the Aβ in the disease’ onset. Physiological Aβ in monomeric form is benign, but by an unknown mechanism it aggregates and becomes neurotoxic [8].”
From previous studies, we have evidenced that APP (amyloid precursor protein) and Aβ protein demonstrated fundamental functions in a variety of central nervous processes in the treatment for AD . Therefore, we did not focus all the pathogenic hallmarks of AD. We highlight the selctive molecules that have shown the inhibition of Aβ levels and for the disease modifying treatments for AD (BACE1 inhibitors). We did not cover the interaction of fluorinated molecules with tau protein as we were focusing on highlighting the role of flourine-containing compounds as anti-amyloid agent. Similarly, we did not cover all the pathofenic fragments of APP. As our main goal is to highlight the importance of fluorine and suggest a new approach in combination with nanotechnology.
Comment 4:
The writing style needs improvement. For example, “The complete disappearance of monomeric NMR signals were observed after 14 h in case of non-fluorinated peptide whereas the peak intensity did not decrease until 0 in presence of fluorinated amino acid indicated the presence of monomeric Aβ42 even after 20 h of incubation and slow kinetic depletion rate.” Such a sentence is hard to understand.
We are grateful for the observation made and this sentence has been rewritten.
“The complete disappearance of monomeric NMR signals was observed after 14 h in case of non-fluorinated peptide. Whereas monomeric Aβ42 was detected even after 20 h incubation.”
Comment 5:
Please proofread the paper and check grammar issues.
Thanks for the comment. The authors proofread the manuscript and made improvements. The changes are highlighted.
Comment 6:
Table 1, the authors list “therapeutic effect”, but the content is hard to understand, and I think “used as a model to understand the…” is not a therapeutic effect. I would also suggest removing the column “experiment”; for the column “models used”, it is hard to understand when the authors just put “ Aβ42 peptide sequence” or “MDRI-MDCK cells”. Similar questions applied to Table 2. Furthermore, the format should be consistent. In table 2, the authors sometimes use present tense, such as “inhibit BACE1 activity”, sometimes used past tense, such as “ Decreased the Aβ levels in CSF”, and sometimes use nouns “Potent reduction of amyloid levels”, please keep consistent.
We totally agree with the reviewer’s comment. In fact, the previous table could induce confusion on readers. Both tables were re-written in present tense. Taking your suggestion, “experiments” column were removed from all the tables. The changes are highlighted in the revised manuscript.
Comment 7:
I suggest the authors discuss the safety and toxicity of fluorinated molecules and nanotechnology when applied alone or in combination for treating Alzheimer’s disease.
We are grateful for the observation made and we totally agree with the referee’s comment. Information about the safety and toxicity of fluorinated molecules were included in the section 2.
Lines 103-112:
“However, unsuitable fluorine positioning in a bio-active compound or a drug molecule can cause stability and toxicity issues. The concern regarding the toxicity of fluorinated molecules has drawn attention to medicinal chemists in the last few years due to the instability of C-F bond in the presence of a nucleophile or drug-metabolizing enzymes [46]. There is a clinical evidence that shows the prolong use of antifungal voriconazole can increase plasma fluoride levels and lead to painful periostitis/exostoses in patients [47].
With the discovery of more innovative fluorinated molecules, medicinal chemists and other researchers are equipped with more powerful tools to recognize the potential liability of fluorinated molecules/drugs and play around in placing fluorine atoms at different position for tackling such issues [46].”
Round 2
Reviewer 2 Report
The concerns are addressed by the authors.